# Aluminum-Immobilizing Rhizobacteria Modulate Root Exudation and Nutrient Uptake and Increase Aluminum Tolerance of Pea Mutant E107 (*brz*)

**DOI:** 10.3390/plants12122334

**Published:** 2023-06-15

**Authors:** Andrey A. Belimov, Alexander I. Shaposhnikov, Tatiana S. Azarova, Oleg S. Yuzikhin, Edgar A. Sekste, Vera I. Safronova, Igor A. Tikhonovich

**Affiliations:** 1All-Russia Research Institute for Agricultural Microbiology, Podbelskogo sh. 3, Pushkin, 196608 Saint-Petersburg, Russia; ai-shaposhnikov@mail.ru (A.I.S.); tatjana-aza@yandex.ru (T.S.A.); yuzikhin@gmail.com (O.S.Y.); sekste_edgar@mail.ru (E.A.S.); v.safronova@rambler.ru (V.I.S.); igor.tikhonovich49@mail.ru (I.A.T.); 2Department of Biology, Saint-Petersburg State University, University Embankment, 199034 Saint-Petersburg, Russia

**Keywords:** aluminum tolerance, *Cupriavidus*, immobilization, pea, PGPRs, rhizosphere, root exudation, symbiosis

## Abstract

It is well known that plant-growth-promoting rhizobacteria (PGPRs) increase the tolerance of plants to abiotic stresses; however, the counteraction of Al toxicity has received little attention. The effects of specially selected Al-tolerant and Al-immobilizing microorganisms were investigated using pea cultivar Sparkle and its Al-sensitive mutant E107 (*brz*). The strain *Cupriavidus* sp. D39 was the most-efficient in the growth promotion of hydroponically grown peas treated with 80 µM AlCl_3_, increasing the plant biomass of Sparkle by 20% and of E107 (*brz*) by two-times. This strain immobilized Al in the nutrient solution and decreased its concentration in E107 (*brz*) roots. The mutant showed upregulated exudation of organic acids, amino acids, and sugars in the absence or presence of Al as compared with Sparkle, and in most cases, the Al treatment stimulated exudation. Bacteria utilized root exudates and more actively colonized the root surface of E107 (*brz*). The exudation of tryptophan and the production of IAA by *Cupriavidus* sp. D39 in the root zone of the Al-treated mutant were observed. Aluminum disturbed the concentrations of nutrients in plants, but inoculation with *Cupriavidus* sp. D39 partially restored such negative effects. Thus, the E107 (*brz*) mutant is a useful tool for studying the mechanisms of plant–microbe interactions, and PGPR plays an important role in protecting plants against Al toxicity.

## 1. Introduction

Aluminum (Al) is a major negative factor in acid soils for the responsible growth inhibition of agricultural crops due to the disturbances of plant metabolism, the induction of oxidative stress, and a decrease in root development, nutrient uptake, and photosynthesis [1,2]. Plants counteract Al toxicity through increasing the rhizosphere pH, complexation by root exudates, production of mucilage and siderophores, detoxification within root tissues, and efflux from roots [3,4,5]. Pea (*Pisum sativum* L.), being a very important agricultural crop, is a relatively Al-sensitive species [6,7,8]. A number of negative effects of Al on pea have been described, namely: inhibition of seed germination [9] and root elongation [7,10,11], induction of oxidative stress [9,10,11,12], a decrease in root and/or shoot biomass [6,8,9,13], and diminution of chlorophyll and nitrogen content in leaves [9]. However, pea varieties significantly differ in Al tolerance; thus, more-tolerant or -sensitive genotypes have been identified [7,8,13,14] and used to understand the mechanisms of Al tolerance. In particular, the counteraction of Al-induced oxidative stress occurs due to the high activity of antioxidative enzymes and the formation of Al-tolerant regions in pea roots, leading to the promotion of the elongation of the central cylinder during recovery from Al toxicity [10,11,12]. An increased formation of pectins in Al-treated pea roots [15] and nitrogen-fixing nodules [16] contributes to the immobilization of Al in plant cell walls. A comparison of 106 pea genotypes revealed the important role of Al precipitation in the root zone via alkalization of the rhizosphere and the maintenance of nutrient homeostasis for the counteraction of toxic effects on plant growth [13].

Plant-growth-promoting rhizobacteria (PGPRs) are well-known symbiotic microorganisms capable of increasing the adaptation of plants to various abiotic stress factors including heavy metal and potassium (soil salinity) toxicity [17,18,19,20,21,22]. However, the role of PGPRs in plant tolerance to Al toxicity and acid soils has received little attention. It was shown that the Al-tolerant associative nitrogen fixer *Azospirillum brasilense* improved the growth of rice [23] and finger millet [24] cultivated in acid soils. An increased Al tolerance of barley inoculated with nitrogen-fixing *Azospirillum lipoferum* 137 and auxin-producing *Flavobacterium* sp. L30 has been described [25]. Al-tolerant strain *Viridibacillus arenosi* IHBB7171, producing auxins and having 1-aminocyclopropane-1-carboxylate (ACC) deaminase activity, stimulated pea growth [26]. Positive effects on maize growth such as root elongation and decreased Al content in roots was observed after inoculation with P-solubilizing *Burkholderia* sp. [27] in the presence of toxic Al concentrations. However, the mechanisms of the anti-stress effects of these bacteria have not been investigated in detail.

Recently, we showed that the biocontrol PGPR *Pseudomonas fluorescens* SPB2137 [28] produced auxins, contained ACC deaminase, immobilized Al, and excluded this toxicant from the root zone of various pea varieties [29]. However, *Ps. fluorescens* SPB2137 was relatively sensitive to Al and exerted little positive effect on the growth of Al-treated plants. It is known that Al is toxic for various bacteria, which manifests itself mainly in the competition with nutrient elements, binding to DNA and enzymes, and disturbance to cell wall and membrane permeability [30]. The mechanisms of Al tolerance in bacteria are in many ways similar to those of plants and include the release of chelating organic compounds and siderophores [30,31,32], alkalization of the medium [33], and intracellular accumulation and immobilization [34]. Therefore, microorganisms (bacteria and yeasts) were subsequently specially selected from the rhizosphere of pea cultivated in acid soils for the traits of resistance and immobilization of Al [35]. The most-promising Al-immobilizing strains assigned to various taxonomic groups were the object of this study.

An effective approach to studying plant–microbe interactions under stressful conditions caused by metal toxicity is the use of plant mutants based on resistance traits [36,37]. With regard to Al toxicity, a plant genetic model based on the pea variety Sparkle and its Al-sensitive mutant E107 (*brz*), which actively acidifies the rhizosphere and accumulates Al in the roots and shoots [38], is of particular interest. Initially, the pleiotropic E107 (*brz*) mutant was characterized by low nodulation and Fe hyper-accumulation, accompanied by necrotic spots on the leaves, these traits being controlled by a single recessive gene, *brz* [39].

The purpose of the present work was to evaluate the beneficial role of specifically selected Al-immobilizing PGPR strains in the adaptation of plants to this toxic metal using pea mutant E107 (*brz*) to substantiate the possibilities and potential of using PGPRs as bioinoculants under the conditions of acidic soils with a high Al content.

## 2. Results

### 2.1. Plant Growth

To select the most-efficient strain, two preliminary hydroponic experiments were performed where all the plants were grown in the presence of 80 µM AlCl_3_ only. All the studied strains were applied in the first experiment for the comparison of their effect on plant growth (Table 1). Strain *Cupriavidus* sp. D39 increased the root and shoot biomass of Sparkle and the root biomass of the E107 (*brz*) mutant (Table 1). The root growth of Sparkle improved after inoculation with the yeast *Rhodotorula* sp. AL1. Strains *Paraburkholderia* sp. A35 and *Paraburkholderia sediminicola* C06 increased shoot biomass, whereas the effect of *Herbaspirillum* sp. E52 was negative. In the second experiment, three of the more efficient strains were compared (Table 1). A significant increase of the root and shoot biomass of both pea genotypes inoculated with *Cupriavidus* sp. D39 was observed. Inoculation with *P. sediminicola* C06 or *Paraburkholderia* sp. A35 increased the root biomass of Sparkle or E107 (*brz*), respectively. Based on these results, strain *Cupriavidus* sp. D39 was chosen for further experiments.

Next, three experiments comparing plant growth response to both treatment with Al and inoculation with *Cupriavidus* sp. D39 were performed. The root and shoot biomass of Al-treated E107 (*brz*) decreased, whereas Sparkle was tolerant to this Al concentration (Figure 1). Inoculation increased the biomass of the roots (Figure 1a) and shoots (Figure 1b) of both pea genotypes. The positive effect of *Cupriavidus* sp. D39 on plant growth was more pronounced (about two-times) on the E107 (*brz*) mutant, as compared to the wild-type Sparkle (about 20%).

### 2.2. Properties of Cupriavidus sp. D39

Strain *Cupriavidus* sp. D39 did not produce siderophores and did not solubilize phosphates. This strain was able to utilize the majority of the tested substances such as organic acids, amino acids, and sugars (Appendix A). The phytohormone salicylic acid (SA) was actively utilized, but indole-3-acetic acid (IAA) was not utilized by *Cupriavidus* sp. D39 (Appendix A).

### 2.3. Distribution of Al in Hydroponic System

At the end of experiments, the presence of *Cupriavidus* sp. D39 decreased the Al concentrations in the nutrient solutions where both pea genotypes were cultivated (Figure 2a). Mutant E107 (*brz*) had an elevated Al concentration in the roots, but inoculation decreased it to the level of Sparkle (Figure 2b). The shoot Al concentration was similar in both pea genotypes and not affected by *Cupriavidus* sp. D39 (Figure 2c). No Al was detected in the untreated plants.

Accumulation of Al by E107 (*brz*) was lower in the roots (Figure 3a), but similar in the shoots (Figure 3b), as compared to Sparkle. Inoculation decreased the total amount of Al in the solution by two-times (Figure 3c). Based on these results, the calculation showed that the total Al in the residue was greater in pots with E107 (*brz*) and increased with *Cupriavidus* sp. D39 (Figure 3d). 

The mutant Al E107 (*brz*) acidified the nutrient solution, particularly when Al was supplemented, but *Cupriavidus* sp. D39 increased the pH of the Al-treated mutant plants (Figure 2d).

### 2.4. Presence of Cupriavidus sp. D39 in Hydroponic System

The number of *Cupriavidus* sp. D39 in the nutrient solution with the E107 (*brz*) mutant was about 2.5-times higher than that of Sparkle and even increased in the presence of Al (Figure 4a). A tendency for a 49% increased bacteria number in the solution of Al-treated Sparkle (t = 2.7; *p* = 0.013; *n* = 12) as compared to the control plants was observed. The roots of both pea genotypes were actively colonized by *Cupriavidus* sp. D39, and the bacteria number increased by about four-times in the presence of Al (Figure 4b). 

The maximal colonization was observed on the Al-treated roots of E107 (*brz*). Single cells and small micro-colonies were observed on the root surface of both pea genotypes grown in the absence of Al in the solution (Figure 5a,b). Relatively large colonies were found on the Al-treated roots of Sparkle (Figure 5c), and particularly on the roots of E107 (*brz*) mutant (Figure 5d).

### 2.5. Root Exudation

Nine organic acids were detected in the root exudates of both pea genotypes (Figure 6).

Mutant E107 (*brz*) showed increased exudation of fumarate (Figure 6c), lactate (Figure 6d), malate (Figure 6e), pyruvate (Figure 6h), and succinate (Figure 6i) by the uninoculated control and Al-treated plants. Al-treated E107 (*brz*) plants also had increased exudation of acetate (Figure 6a) and pyroglutamate (Figure 6g). In the presence of Al, the increased exudation of citrate (Figure 6b), propionate (Figure 6f), and succinate (Figure 6i) by Sparkle was observed, whereas E107 (*brz*) plants had increased exudation of fumarate (Figure 6c), lactate (Figure 6d), malate (Figure 6e), and succinate (Figure 6i). Inoculation with *Cupriavidus* sp. D39 decreased the amount of acetate (Figure 6a), citrate (Figure 6b), lactate (Figure 6d), propionate (Figure 6f), pyroglutamate (Figure 6g), and pyruvate (Figure 6h) in the control and/or Al-treated solution of Sparkle. The amount of these organic acids, as well as malate (Figure 6e) and succinate (Figure 6i) was also significantly lower in the solutions where the inoculated E107 (*brz*) mutant was cultivated. The total exudation of organic acids by the E107 (*brz*) mutant was about three-times higher compared to Sparkle in the absence of *Cupriavidus* sp. D39 (Figure 7a). However, the inoculated plants had a 3–5-times lesser amount of exuded organic acids, particularly for the E107 (*brz*) mutant (Figure 7a). 

Mutant E107 (*brz*) exuded more actively many of the amino acids, such as isoleucine (Figure 8f), leucine (Figure 8g), lysine (Figure 8h), methionine (Figure 8i), phenylalanine (Figure 8j), proline (Figure 8k), serine (Figure 8l), tryptophan (Figure 8n), and tyrosine (Figure 8o), as well as alanine, glutamic acid, and valine (Appendix A). As a rule, genotypic differences were retained in the presence of Al in the nutrient solution (Figure 8, Appendix A).

The treatment of Sparkle with Al decreased the exudation of nine out of fifteen amino acids, as shown in Figure 8, and of alanine and aspartic acid (Appendix A). The Al-treated mutant roots exuded less histidine (Figure 8e), isoleucine (Figure 8f), leucine (Figure 8g), lysine (Figure 8h), proline (Figure 8k), alanine, and valine (Appendix A). Moreover, the exudation by E107 (*brz*) of arginine (Figure 8a), cysteine (Figure 8b), GABA (Figure 8c), glycine (Figure 8d), tryptophan (Figure 8n), and ornithine (Appendix A) increased with the Al treatment. In the absence of Al, inoculation with *Cupriavidus* sp. D39 decreased the amount of many amino acids, except threonine (Figure 8m), glutamic acid, and ornithine (Appendix A) in Sparkle exudates, as well as arginine (Figure 8a), GABA (Figure 8c), and threonine (Figure 8m) in E107 (*brz*) exudates. In line with this, the decrease in the amount of many amino acids was observed for Al-treated plants, except proline (Figure 8k), threonine (Figure 8m), and tyrosine (Figure 8p) exuded by Sparkle, as well as arginine (Figure 8a), GABA (Figure 8c), threonine (Figure 8m), aspartic acid, glutamic acid, and ornithine (Appendix A) exuded by E107 (*brz*). In general, the treatment with Al decreased the total exudation of amino acids by Sparkle (Figure 7b). The inoculation with *Cupriavidus* sp. D39 decreased the amount of amino acids exuded by both genotypes in the absence of Al and of those exuded only by E107 (*brz*) in the presence of Al (Figure 7b).

Only E107 (*brz*) roots exuded glucose (Figure 9c), and the exudation of ribose (Figure 9d) and sucrose (Figure 9e) was significantly higher by E107 (*brz*) roots. The treatment with Al induced the exudation of arabinose (Figure 9a) and increased the exudation of fructose (Figure 9b) by both pea genotypes, but decreased the exudation of glucose (Figure 9c), ribose (Figure 9d), and sucrose (Figure 9e) by E107 (*brz*). As a rule, the effects of *Cupriavidus* sp. D39 on the amount of various sugars was negative on the control and/or Al-treated plants, particularly on the E107 (*brz*) mutant (Figure 9). 

The total amount of exuded sugars was about two-times greater for E107 (*brz*) as compared with Sparkle, whereas inoculation with *Cupriavidus* sp. D39 dramatically decreased the total amount of sugars in pea genotypes (Figure 7c). 

Phytohormone IAA was detected in the solution where the inoculated E107 (*brz*) mutant was cultivated with and without the addition of Al (Figure 10a). A relatively low amount of IAA was also found in the solution of Al-treated Sparkle inoculated with *Cupriavidus* sp. D39 (Figure 10a). Mutant E107 (*brz*) exuded three-times more SA, and treatment with Al significantly inhibited the exudation of this phytohormone (Figure 10b). The inoculated treatments of both pea genotypes showed trace amounts of SA only.

### 2.6. Nutrient Uptake by Plants

The treatment with Al increased the concentration of Fe in E107 (*brz*) roots and the concentration of P in roots of both uninoculated pea genotypes (Table 2). Decreased K, Mg, and S concentrations were found in uninoculated Al-treated E107 (*brz*) roots, whereas the root Mg concentration of E107 (*brz*) also decreased in the presence of *Cupriavidus* sp. D39. The inoculation with *Cupriavidus* sp. D39 increased the root Mg and Mn concentrations in both pea genotypes and the K and S concentrations of the inoculated E107 (*brz*) (Table 2). Control mutant roots had increased Mn, P, and Zn concentrations. 

Al-treated plants had decreased concentrations of Ca and Mg in the shoots of the inoculated Sparkle and uninoculated E107 (*brz*), respectively (Table 3). Inoculation with *Cupriavidus* sp. D39 increased the Ca concentration in the shoots of both pea genotypes, as well as the Fe and Mg concentrations in the Al-treated Sparkle and E107 (*brz*), respectively (Table 3).

## 3. Discussion

### 3.1. Comparison of Al-Immobilizing Strains for Plant Growth Promotion

In the first stage of this research, five recently isolated Al-immobilizing strains [35] were compared for their ability to promote the growth of the pea Sparkle and the E107 (*brz*) mutant in the presence of a toxic Al concentration. The effect of bacteria and yeasts varied from slightly negative to positive depending on the strain and pea genotype (Table 1). The maximum plant biomass gains were obtained after inoculations with *Cupriavidus* sp. D39 (Table 1), providing the reason for choosing this strain for further detailed study.

Probably, apart from the immobilization of Al in the root zone, these strains could affect plant growth by other mechanisms. All the tested bacterial strains produced IAA and utilized ACC as a nitrogen source, suggesting the presence of ACC deaminase activity [35]. It was previously suggested that these traits could play an important role in plant growth promotion under stress conditions caused by Al toxicity. In particular, *Ps. fluorescens* SPB2137 produced auxins, had ACC deaminase activity, and increased Al tolerance of pea [29]. Al-tolerant strain *Viridibacillus arenosi* IHBB7171 promoted the growth of pea plants, probably due to producing auxins and the ACC deaminase activity [26]. The increased biomass of ryegrass grown in a volcanic soil rich in Al was obtained after inoculation with IAA-producing and ACC-utilizing *Klebsiella* sp., *Stenotrophomonas* sp., *Serratia* sp., and *Enterobacter* sp. [40]. A decreased Al toxicity on the growth of mung bean inoculated with *Bacillus megaterium* CAM12 and *Pantoea agglomerans* CAH6 [41] or lettuce inoculated with *Curtobacterium herbarum* CAH5 [42] and *Rhodotorula mucilaginosa* CAM4 [43] has also been described. In addition, *Cupriavidus* sp. D39 can fix atmospheric nitrogen [35], having a set of *fix* genes in its genome [44].

### 3.2. Plant Biomass and Al Uptake

The pronounced decrease in the biomass of the Al-treated E107 (*brz*) mutant as compared to Sparkle was an expected result, since this mutant was initially characterized as a Al-sensitive genotype due its ability to acidify the rhizosphere and accumulate Al [38]. The root and shoot growth promotion caused by *Cupriavidus* sp. D39 on Al-treated plants was significantly higher for E107 (*brz*) (Figure 1). This result is in line with our previous finding, showing higher growth promotion caused by *Ps. fluorescens* SPB2137 on Al-sensitive pea varieties as compared to Al-tolerant varieties [29]. The effect observed in this study presents original indirect evidence of the bacterial contribution to plant growth promotion by reducing the availability of Al for inoculated plants, since the genetic backgrounds of Sparkle and E107 (*brz*) are the same. This was also confirmed by the decrease in the concentration of Al in the nutrient solution of both pea genotypes and in the roots of the Al-accumulating E107 mutant (*brz*) only (Figure 1), as well as by the increased Al accumulation in the residue (Figure 3d). The ability of *Cupriavidus* sp. D39 to alkalize the nutrient solution could facilitate the precipitation of Al and alleviate its toxic effect on the roots. 

There have been several reports demonstrating reduced Al uptake by plants inoculated with PGPRs [33,34,35]. However, the mechanisms of such an effect have not been studied enough. A decreased Al concentration in the batch culture of *C. herbarum* CAH5 was associated with the accumulation of P in the residue, suggesting the formation of insoluble phosphates [42]. PGPR strain *Burkholderia* sp. increased the root length of maize and decreased Al accumulation in roots as a result of the binding of Al ions by soil phosphates [27]. The binding toxic Al with phosphates solubilized from soil by several PGPR strains has been pointed out as a possible mechanism for the growth promotion of ryegrass [40]. However, we found here that strain *Cupriavidus* sp. D39 did not solubilize phosphates in vitro and even decreased the water-soluble P concentration in soil, accompanied by its alkalization [35]. It has also been proposed that PGPRs produce siderophores, leading to the formation of Al^3+^–siderophore complexes, which are hardly available for plants [40]. This property is unlikely to apply to *Cupriavidus* sp. D39, since no siderophores were found in its culture fluid. 

The presence of high Al concentrations in the growth medium of *B. megaterium* CAM12 and *P. agglomerans* CAH6 induced the production of exopolysaccharides, contributing to biofilm formation [41]. Strain *Ps. fluorescens* ATCC 13525 responded to Al toxicity by the biosynthesis of phospholipids, leading to the formation of residue containing Al [45,46]. The treatment with Al of *Ps. fluorescens* SPB2137 in batch culture induced the development of clumps considered as biofilm-like structures containing cells, exopolysaccharides, phospholipids, and Al phosphates [29]. The present report demonstrated the formation of characteristic micro-colonies of *Cupriavidus* sp. D39 on Al-treated pea roots (Figure 5c,d), which looked like the clumps previously observed in *Ps. fluorescens* SPB2137 [29]. We propose that such clumps could be involved in the protection of roots from Al toxicity by *Cupriavidus* sp. D39. 

Another important observation was that *Cupriavidus* sp. D39, having a high Al tolerance, very actively colonized the pea roots, particularly in the presence of Al (Figure 4b). In contrast, pea root colonization by *Ps. fluorescens* SPB2137 was inhibited by the Al treatment, and the bacteria number on the roots was about 100-times less [29] as compared with that observed here for *Cupriavidus* sp. D39 (Figure 4b). 

### 3.3. Root Exudation

For the first time, root exudation of organic compounds by the E107 (*brz*) mutant was described, and an increased exudation of organic acids, amino acids, and sugars as compared to the wild-type Sparkle was revealed. Moreover, the treatment with Al stimulated exudation, particularly by the mutant plants. The E107 (*brz*) mutant not only had an increased release of protons [38,47], but also an abnormal efflux of various low-molecular-weight organic compounds. This makes the E107 (*brz*) mutant a unique plant for studying plant physiology and plant–microbe interactions, taking into account that these compounds play an important role in plant metabolism and nutrition, as well as serve as a major nutrient source for symbiotic microorganisms. 

The observed upregulated efflux of various plant metabolites and photosynthates by the E107 (*brz*) roots could be responsible for its impaired growth reported earlier [38] and is consistent with our results (Figure 1). The increased exudation of organic acids by the roots of the E107 (*brz*) mutant did not help counteract the Al toxicity. It is known that the exuded organic acids are involved in the complexation of Al in the root zone, and this phenomenon is considered an important mechanism of Al tolerance in many plant species [4]. However, there was no correlation between the intensity of organic acid exudation and the growth response to Al toxicity of 11 pea genotypes [13]. The obtained results suggested that, although Al toxicity induced the exudation of organic acids by pea roots, their protective role in the Al tolerance of this species is ambiguous. 

Previously, it was shown that treatment with SA reduced oxidative stress, the accumulation of Al in shoots, and stimulated the growth of Al-treated rice [48], alfalfa [49], soybean [50], and *Cassia tora* [51]. Treatment with Al increased the endogenous SA concentration in the roots of soybean [52] and induced the expression of genes related to SA biosynthesis in *Vitis quinquangularis* [53]. Here, the treatment with Al inhibited the exudation of SA by both pea genotypes, whereas *Cupriavidus* sp. D39 utilized it in vitro (Appendix A) and most probably in the nutrient solution (Figure 10). An intriguing question arises: Can the decrease in SA exudation under the action of Al be a protective response of plants due to its preservation inside root tissues and/or translocation into shoots? The results also suggested that it is difficult to explain the positive effects of *Cupriavidus* sp. D39 on pea growth and Al accumulation due to SA’s modulation. 

Aluminum affects biosynthesis and disturbs the balance of endogenous amino acids in plants [3,54,55]. The roots of wheat seedlings responded to Al toxicity by increased exudation of many proteinogenic amino acids [56]. The exudation of proteinogenic amino acids by four pea genotypes was also increased in Al-treated plants [29]. At the same time, a decreased exudation of asparagine, isoleucine, and glutamic acid was observed in Al-treated trifoliate orange [54]. In the present study, the Al treatment inhibited amino acid exudation by the cultivar Sparkle (Figure 7b and Figure 8). The effect of Al on the E107 (*brz*) mutant varied from positive to negative depending on the amino acid (Figure 8), resulting in a similar exudation of the total amount of amino acids of both pea genotypes (Figure 7b). The results suggested high genotypic variability in amino acid exudation and disturbance of this trait in the E107 (*brz*) mutant.

The protective action of amino acids against Al toxicity has been repeatedly reported: arginine is involved in the biosynthesis of putrescine, leading to decreased Al retention in the cell walls of wheat [57]; cysteine increased the activity of glutathione [58]; GABA induced the antioxidant system in barley [59] and *Arabidopsis thaliana* [60]; glycine played a stress alleviating role as a component of glycine betaine in rice [61] and glycine-rich proteins in *A. thaliana* [62]; phenylalanine was involved in the biosynthesis of phenylpropanoids and flavonoids with defense functions in alfalfa [63,64]; proline protected root cells as an osmolyte [55,61,65]; threonine activated H^+^-ATPase in soybean [66]. Four of these amino acids (Arg, Cys, GABA, and Gly) were exuded more by the E107 (*brz*) mutant treated with Al (Figure 8a–d). Therefore, it is possible that the disturbance of amino acid exudation and the loss of these substances from the roots can negatively affect the adaptation of E107 (*brz*) to the toxic Al. 

The decreased amount of amino acids in the nutrient solution inoculated with *Cupriavidus* sp. D39 was probably due to their utilization as a nitrogen or carbon source, since these compounds were nutrients for the bacteria in vitro (Appendix A). It is important to pay attention to the increased exudation of tryptophan by E107 (*brz*), especially in the presence of Al, and its absence in the inoculated solution (Figure 8o). The increased exudation of tryptophan by pea roots treated with Al was observed previously [29], and the present results confirmed this observation. PGPRs use tryptophan for the biosynthesis of auxins, which induces the expression of genes involved in the plant’s Al tolerance [67], increased root exudation of organic acids [68,69], and promoted the growth of Al-treated plants [70,71,72,73,74]. However, the overproduction of auxins due to Al toxicity in *A. thaliana* roots inhibited root elongation [75]. On the other hand, the treatment of alfalfa with Al decreased the IAA concentration in apical buds and root tips, resulting in root growth inhibition [73]. This means that the optimal auxin concentration is needed to counteract the toxic effects of Al on plant growth. It could be proposed that the auxin-producing *Cupriavidus* sp. D39 [35] could be used to explore the intensive exudation of tryptophan by the E107 (*brz*) mutant (Figure 8o) for IAA production (Figure 10a), leading to growth promotion (Figure 1). Recently, we suggested such a mechanism for pea growth promotion by IAA-producing *Ps. fluorescens* SPB2137 [29]. The growth promotion of Al-treated maize seedlings by *Ps. fluorescens* 002 was proposed due to its auxin production [76]. Here, this hypothesis was confirmed by the disappearance of tryptophan and the presence of IAA in the nutrient solution of the inoculated E107 (*brz*) mutant. 

The influence of Al stress on the exudation of sugars by plant roots has received little attention. The increased exudation of glucose by wheat [56] and fructose, glucose, and ribose by pea [29] was previously described for Al-treated seedlings. Intra-species variation in sugar exudation was also described for pea [29] and soybean [77]. Here, quantitative and qualitative differences in the exudation of sugars between the E107 (*brz*) mutant and its parental cultivar Sparkle in the presence and absence of toxic Al were revealed (Figure 9). As with organic acids and amino acids, excessive exudation of photosynthates in the form of sugars by the E107 (*brz*) mutant could be negatively manifested in the adaptation to Al stress. On the other hand, this trait could support active root colonization by *Cupriavidus* sp. D39 (Figure 4b), since the strain used these substances as nutrients (Appendix A) and decreased their amount in the solution (Figure 9), enhancing the plant-growth-promoting effects, particularly on Al-treated plants. 

### 3.4. Nutrient Uptake by Plants

Mutant E107 (*brz*) possessed low tolerance, but increased accumulation of Fe [39] and Al [38] due to the acidification of the rhizosphere. The previously hydroponically cultivated E107 (*brz*) had elevated shoot Mn and Zn concentrations when growing in Fe-deficient solution, but the supplement with 10 µM Fe-citrate further increased the root Fe and the shoot Fe, Mn, Zn, and Cu concentrations [39,47]. Experiments with vermiculite and soil cultures revealed the ability of the E107 (*brz*) mutant to actively accumulate Ca, Fe, K, Mg, and Mn in the leaves [47]. In this study, genotypic differences between Sparkle and the E107 (*brz*) mutant were also found in the root concentrations of Mn, P, and Zn (Table 2) and in the shoot concentration of Ca (Table 3) in the absence of Al in the nutrient solution. The treatment with Al significantly affected the genotypic differences in the root concentrations of Fe, K, Mg, S, and Zn, as well as in shoot concentrations of Ca and Mg. This new information about the nutrient uptake by the E107 (*brz*) mutant is important for its detailed characterization.

The effects of *Cupriavidus* sp. D39 on nutrient uptake were directed towards increasing concentrations of elements in plants and were more pronounced in the Al-treated mutant plants (Table 2 and Table 3). The latter observation was probably due to the large number of bacteria in this variant of the experiment. However, the effect of Al on the uptake of nutrients was, for the most part, negative. In particular, *Cupriavidus* sp. D39 eliminated the negative effects of Al on the concentrations of K, Mg, and S in the mutant roots and of Mg in the mutant shoots. It was reported that the toxic Al inhibits nutrient uptake in various crops [1,2] including pea [78], and the ability of counteracting Al-induced disturbances of nutrient uptake was shown as a principal mechanism for the Al tolerance of this plant species [13]. Moreover, the inoculation of pea with a microbial consortium containing PGPR *Ps. fluorescens* SPB2137 [79] improved the uptake of nutrients from Al-contaminated soil [79]. We propose that *Cupriavidus* sp. D39 contributes to the adaptation of plants to the Al toxicity via the enhancement of the nutrient uptake by the roots and that such a positive effect is more pronounced and more important for Al-sensitive plant genotypes.

## 4. Materials and Methods

### 4.1. Plants

Seed samples of the pea (*Pisum sativum* L.) cultivar Sparkle and its mutant E107 (*brz*) were kindly provided by Prof. Frederique C. Guinel. The seeds were propagated under the same soil and climate conditions in summer under natural light, air moisture, and temperature (experimental greenhouse of ARRIAM, Saint Petersburg, Russia).

### 4.2. Microorganisms

Strains of the Al-immobilizing bacteria *Cupriavidus* sp. D39, *Paraburkholderia* sp. A35, *Paraburkholderia sediminicola* C06, and *Herbaspirillum* sp. E52 and yeast *Rhodotorula* sp. AL1 [28] were obtained from the Russian Collection of Agricultural Microorganisms (RCAM) (St.-Petersburg, Russian Federation; http://www.arriam.ru/kollekciya-kul-tur1/; accessed on 7 July 2021) and maintained on potato dextrose agar (PDA) medium (HiMedia Laboratories Pvt. Ltd., Mumbai, India). The strain *Cupriavidus* sp. D39 was marked with the gene encoding green fluorescent protein (GFP) using the mini-transposon suicide delivery system pAG408 [80] as previously described [81]. The utilization of organic acids, amino acids, sugars, and indole-3-acetic acid (IAA) by *Cupriavidus* sp. D39 was assessed in vitro via the cultivation of bacteria in a mineral liquid medium containing these substances as a sole carbon or nitrogen source [77]. Siderophore production by *Cupriavidus* sp. D39 was determined using a chrome azurol S (CAS) shuttle solution, as described by Schwyn and Neilands [82]. The assay was calibrated by generating a standard curve for samples containing 1 to 100 μM deferoxamine mesylate. The capability of *Cupriavidus* sp. D39 to solubilize phosphates was tested in vitro by detecting the clearing zones developed around the bacterial colonies grown for 5 days on an agar medium supplemented with 500 mg L^−1^ of Ca_3_(PO_4_)_2_, Ca-phytate, Al_3_PO_4_, or Fe_3_PO_4_ [83].

### 4.3. Plant Growth

An original method of growing peas in a hydroponic gnotobiotic system was applied [29]. Briefly, seeds of cv. Sparkle and E107 (*brz*) were surface-sterilized and scarified by treatment with 98% H_2_SO_4_ for 20 min, rinsed with sterile tap water, and germinated in Petri dishes for three days at 25 °C. Then, ten seedlings were transferred to each polypropylene pot (OS140BOX, Duchefa, Netherlands) containing 250 mL of sterile nutrient solution (µM): KNO_3_, 1200; Ca(NO_3_)_2_, 60; MgSO_4_, 250; KCl, 250; CaCl_2_, 60; Fe-tartrate, 12; H_3_BO_3_, 2; MnSO_4_, 1; ZnSO_4_, 3; NaCl, 6; Na_2_MoO_4_, 0.06; CoCl_2_, 0,06; CuCl_2_, 0.06; NiCl_2_, 0,06; pH = 4.7. The nutrient solution was supplemented or not with 80 µM AlCl_3_×6H_2_O and/or inoculated with bacteria in a final concentration of 10^5^ cells mL^−1^. Non-supplemented solution was used as a control treatment. Plants were cultivated for 10 days in a growth chamber (ADAPTIS-A1000, Conviron, Isleham, UK) with 200 µmol of quanta m^−2^ s^−1^, a 12 h photoperiod, and minimum/maximum temperatures of 18 °C/23 °C. Then, the biomass of individual plants was determined, and the pH of the nutrient solution was measured using the pH meter F20 (Mettler-Toledo, Schwerzenbach, Switzerland). Five experiments with one pot per treatment were performed.

### 4.4. Presence of Cupriavidus sp. D39 in Hydroponic System

At the end of experiments, the number of *Cupriavidus* sp. D39 in the nutrient solution was determined by the serial dilution method using PDA medium as described previously [29]. Samples of the roots were picked, and images were taken to estimate the presence of GFP-tagged *Cupriavidus* sp. D39 using a fluorescent microscope (Axio Imager A2, Carl Zeiss, Oberkochen, Germany).

### 4.5. Determination of Root Exudation

The nutrient solution was centrifuged for 15 min at 9000× *g* and 4 °C. The supernatant was vacuum filtered through 0.45 µm filters (Corning, Germany) and concentrated at 45 °C using a rotary vacuum evaporator (BUCHI R-200 (BUCHI, Switzerland)). The concentrate was analyzed for root exudates (organic acids, sugars, amino acids, IAA, and SA) using the UPLC system Waters ACQUITY H-Class (Waters, Milford, MA, USA) as described previously [77,84].

### 4.6. Elemental Analysis

Aliquots of the supernatants were taken for the determination of the Al concentration in nutrient solutions. The roots and shoots were dried at 50 °C, combined into one sample for each pot, and ground to a powder. All samples were digested in a mixture of concentrated HNO_3_ and 38% H_2_O_2_ at 70 °C using the digestion system DigiBlock (LabTech, Sorisole, Italy). The content of Al and the nutrient elements (Ca, Fe, K, Mg, Mn, P, S, and Zn) in the digested samples was determined using an inductively coupled plasma emission spectrometer (ICPE-9000, Shimadzu, Tokyo, Japan), according to the manufacturer’s instructions. At the end of the experiments, a precipitate (residue) formed in the pots, which in previous studies with peas and PGPR was collected and analyzed and contained precipitated Al [36]. In this work, the amount of Al in this residue was calculated as the difference between the amount of Al introduced into the pot and its total amount in the plants and the solution.

### 4.7. Statistical Analysis

The statistical analysis of the data was performed using the software STATISTICA Version 10 (TIBCO Software Inc., Palo Alto, CA, USA). MANOVA analysis with Fisher’s LSD test and Student’s *t* test were used to evaluate differences between the means.

## 5. Conclusions

Al-tolerant and Al-immobilizing bacteria recently selected from the rhizosphere of pea grown in acid soils showed plant-growth-promoting activity for pea variety Sparkle and, particularly, for its Al-sensitive and Al-accumulating mutant, E107 (*brz*). The most-efficient strain, *Cupriavidus* sp. D39, increased the root and shoot biomass of the Al-treated E107 (*brz*) by about two-times. The growth-promoting effect was accompanied by the immobilization of Al and the decrease in the Al concentration in the roots. The E107 (*brz*) mutant showed upregulated exudation of many low-molecular-weight organic compounds (organic acids, amino acids, and sugars) in the absence or presence of a toxic Al concentration. In most cases, the treatment with Al increased exudation, particularly by the mutant plants. On the one hand, the abundant presence of exudates (potential nutrients for bacteria) in the root zone increased the proliferation, root colonization, and probably, metabolic activities of the bacteria, leading to a more intensive growth promotion of E107 (*brz*). This could be confirmed by the exudation of tryptophan and the formation of IAA by *Cupriavidus* sp. D39 in the root zone of the Al-treated mutant. On the other hand, we hypothesized that excessive loss of carbon, nitrogen, and probably, other elements originating from photosynthates and seeds could impair the metabolism and defense responses of E107 (*brz*) to Al stress. The treatment with Al disturbed the concentration of nutrients in the plants, particularly in the mutant roots, but inoculation with *Cupriavidus* sp. D39 partially restored such negative effects. Thus, a pair of cultivar Sparkle and its E107 (*brz*) mutant was a very suitable genetic model for studying the mechanisms of the bacterial effects on plants in the presence of Al toxicity. The Al-immobilizing PGPRs like *Cupriavidus* sp. D39 offer a promise for increasing the tolerance of plants to Al toxicity under acid soil conditions.

## Figures and Tables

**Figure 1 plants-12-02334-f001:**
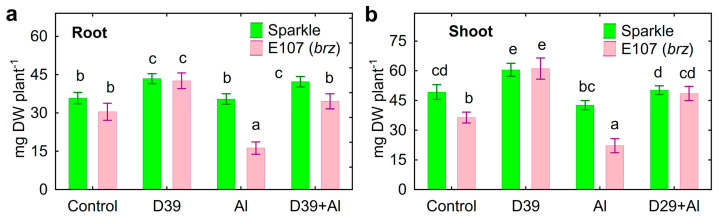
Growth response of roots (**a**) and shoots (**b**) of cultivar Sparkle and E107 (*brz*) mutant to inoculation with *Cupriavidus* sp. D39. Treatments: control—Al-untreated and uninoculated plants, D39—inoculated with *Cupriavidus* sp. D39, Al—treated with 80 µM AlCl_3_, D39 + Al—inoculated with *Cupriavidus* sp. D39 and treated with 80 µM AlCl_3_. Vertical bars show standard errors. Different lowercase letters show significant differences between treatments (least-significant difference test, *p* < 0.05; *n* varied from 25 to 30 depending on the pea genotype and treatment). DW stands for dry weight.

**Figure 2 plants-12-02334-f002:**
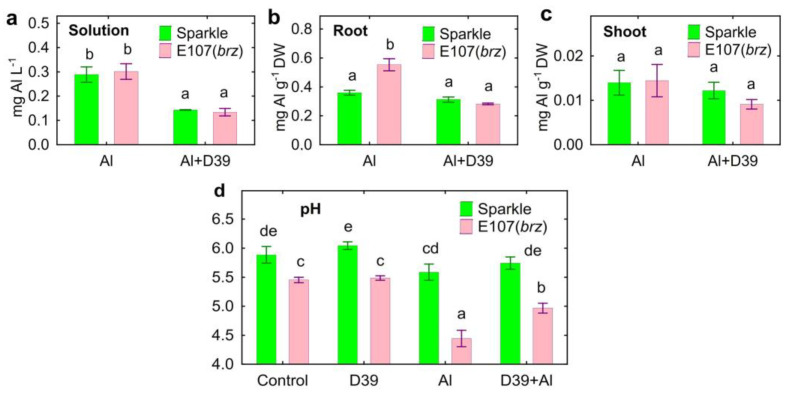
Final concentration of Al in solution (**a**), roots (**b**), and shoots (**c**) and pH of the solution (**d**) of cultivar Sparkle and E107 (*brz*) mutant inoculated with *Cupriavidus* sp. D39. Treatments: control—Al-untreated and uninoculated plants, D39—inoculated with *Cupriavidus* sp. D39, Al—treated with 80 µM AlCl_3_, D39 + Al—inoculated with *Cupriavidus* sp. D39 and treated with 80 µM AlCl_3_. Vertical bars show standard errors. Different lowercase letters show significant differences between treatments (least-significant difference test, *p* < 0.05, *n* = 3). DW stands for dry weight.

**Figure 3 plants-12-02334-f003:**
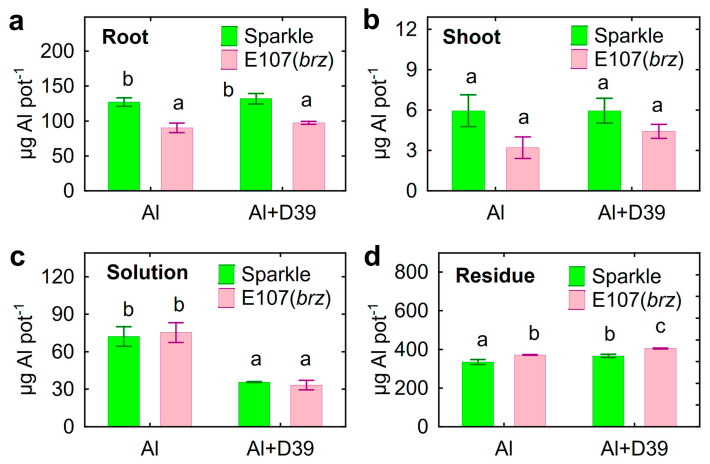
Accumulation of Al in components of hydroponic system: root (**a**), shoot (**b**), solution (**c**), and residue (**d**). Pea genotypes: cultivar Sparkle and E107 (*brz*) mutant. Treatments: Al—uninoculated and treated with 80 µM AlCl_3_, Al + D39—inoculated with *Cupriavidus* sp. D39 and treated with 80 µM AlCl_3_. Vertical bars show standard errors. Different lowercase letters show significant differences between treatments (least-significant difference test, *p* < 0.05, *n* = 3). DW stands for dry weight.

**Figure 4 plants-12-02334-f004:**
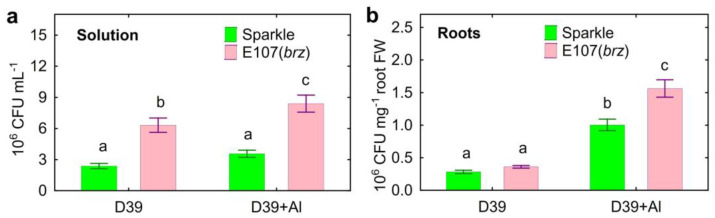
The number of *Cupriavidus* sp. D39 in the nutrient solution (**a**) and on roots (**b**) of cultivar Sparkle and E107 (*brz*) mutant. Treatments: D39—inoculated with *Cupriavidus* sp. D39, D39 + Al—inoculated with *Cupriavidus* sp. D39 and treated with 80 µM AlCl_3_. Vertical bars show standard errors. Different lowercase letters show significant differences between treatments (least-significant difference test, *p* < 0.05, *n* = 12). CFU stands for colony-forming units.

**Figure 5 plants-12-02334-f005:**
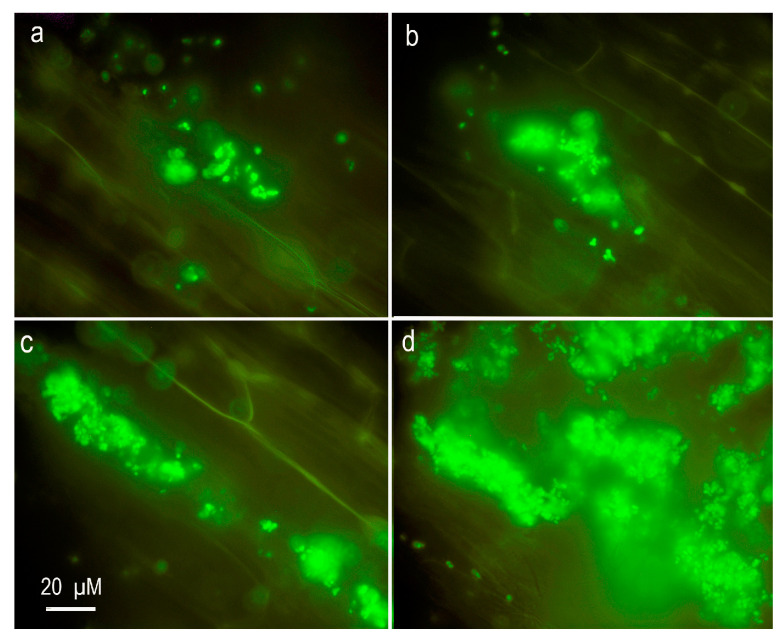
Fluorescence microscopy images of *Cupriavidus* sp. D39 on roots of cultivar Sparkle (**a**,**c**) and E107 (*brz*) mutant (**b**,**d**). The plants were untreated (**a**,**b**) or treated with 80 µM AlCl_3_ (**c**,**d**). The bacteria were tagged with GFP and colored green. Scale bar (20 µm) shown in (**c**) is the same for all images.

**Figure 6 plants-12-02334-f006:**
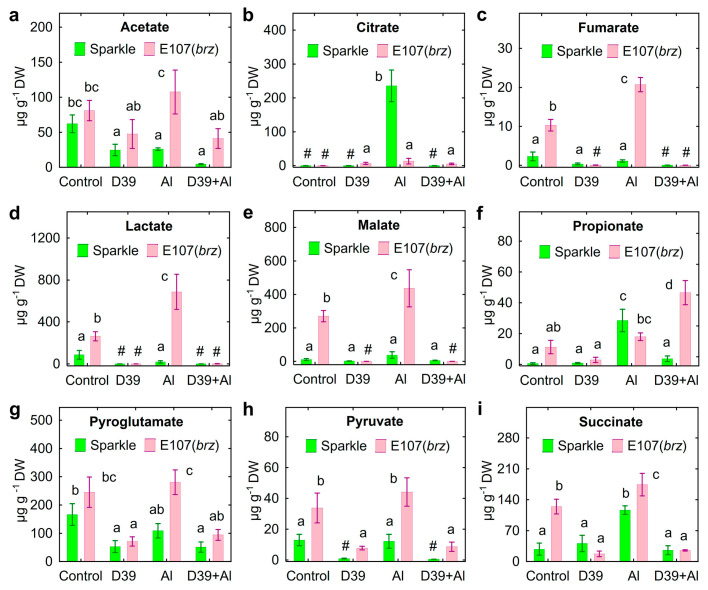
Exudation of organic acids by roots of cultivar Sparkle and E107 (*brz*) mutant inoculated with *Cupriavidus* sp. D39 and treated with aluminum. Organic acids: acetate (**a**), citrate (**b**), fumarate (**c**), lactate (**d**), malate (**e**), propionate (**f**), pyroglutamate (**g**), pyruvate (**h**), succinate (**i**). Treatments: control—Al-untreated and uninoculated plants, D39—inoculated with *Cupriavidus* sp. D39, Al—treated with 80 µM AlCl_3_, D39 + Al—inoculated with *Cupriavidus* sp. D39 and treated with 80 µM AlCl_3_. Vertical bars show standard errors. Different lowercase letters show significant differences between treatments (least-significant difference test, *p* < 0.05, *n* = 3). The # sign means not detected. DW stands for dry weight.

**Figure 7 plants-12-02334-f007:**
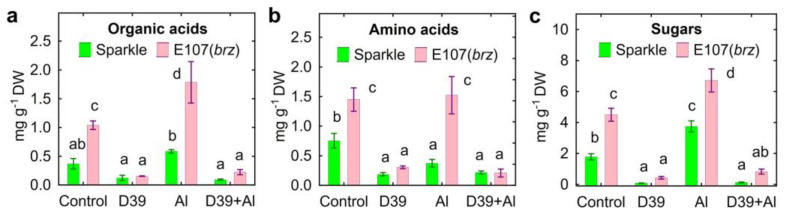
Exudation of total organic acids (**a**), amino acids (**b**), and sugars (**c**) by roots of cultivar Sparkle and E107 (*brz*) mutant inoculated with *Cupriavidus* sp. D39 and treated with aluminum. Treatments: control—Al-untreated and uninoculated plants, D39—inoculated with *Cupriavidus* sp. D39, Al—treated with 80 µM AlCl_3_, D39 + Al—inoculated with *Cupriavidus* sp. D39 and treated with 80 µM AlCl_3_. Vertical bars show standard errors. Different lowercase letters show significant differences between treatments (least-significant difference test, *p* < 0.05, *n* = 3). DW stands for dry weight.

**Figure 8 plants-12-02334-f008:**
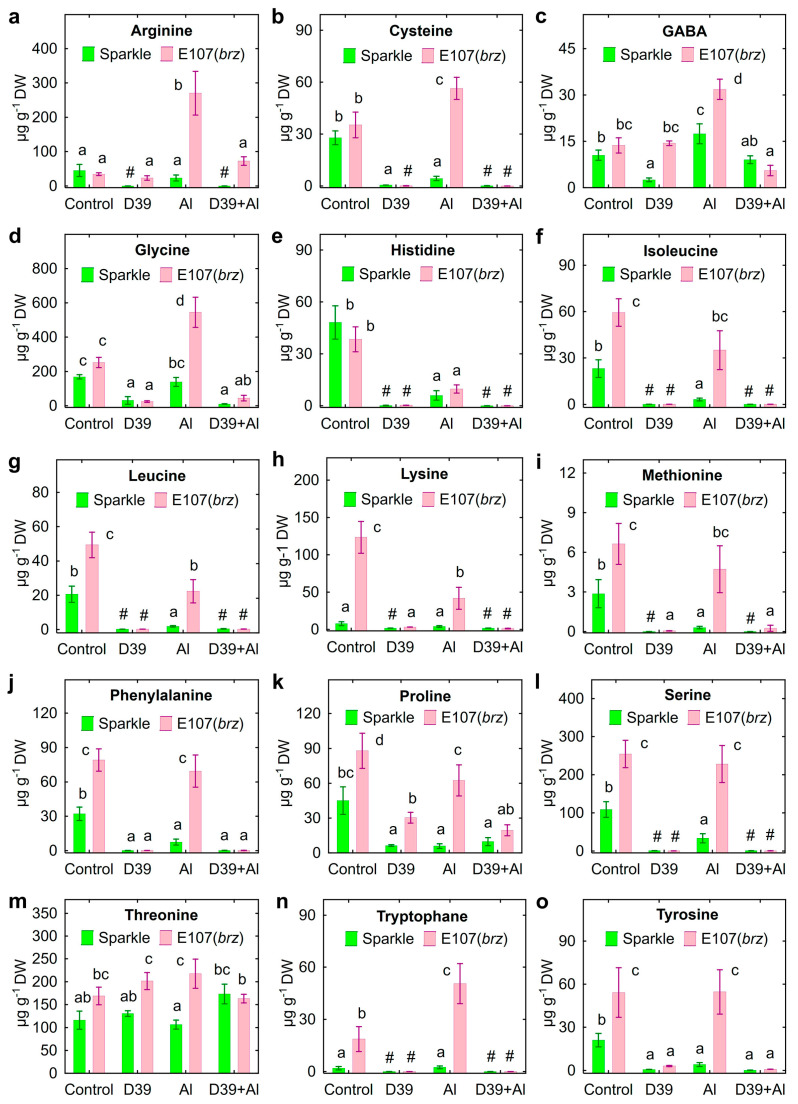
Exudation of amino acids by roots of cultivar Sparkle and E107 (*brz*) mutant inoculated with *Cupriavidus* sp. D39 and treated with aluminum. Amino acids: arginine (**a**), cysteine (**b**), GABA (**c**), glycine (**d**), histidine (**e**), isoleucine (**f**), leucine (**g**), lysine (**h**), methionine (**i**), phenylalanine (**j**), proline (**k**), serine (**l**), threonine (**m**), tryptophane (**n**), tyrosine (**o**). Treatments: control—Al-untreated and uninoculated plants, D39—inoculated with *Cupriavidus* sp. D39, Al—treated with 80 µM AlCl_3_, D39 + Al—inoculated with *Cupriavidus* sp. D39 and treated with 80 µM AlCl_3_. Vertical bars show standard errors. Different lowercase letters show significant differences between treatments (least-significant difference test, *p* < 0.05, *n* = 3). The # sign means not detected. DW stands for dry weight.

**Figure 9 plants-12-02334-f009:**
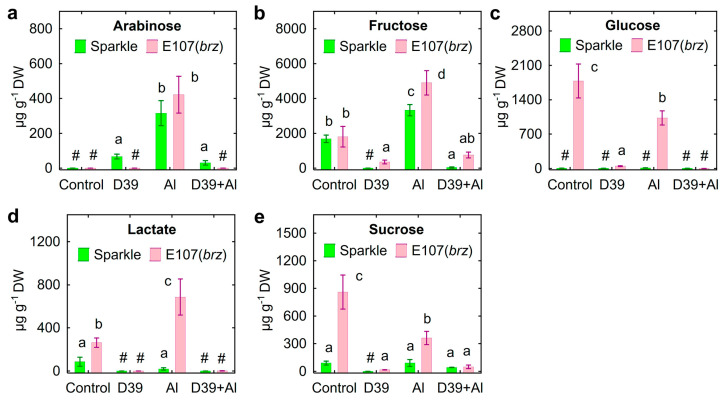
Exudation of sugars by roots of cultivar Sparkle and E107 (*brz*) mutant inoculated with *Cupriavidus* sp. D39 and treated with aluminum. Sugars: arabinose (**a**), fructose (**b**), glucose (**c**), lactate (**d**), sucrose (**e**). Treatments: control—Al-untreated and uninoculated plants, D39—inoculated with *Cupriavidus* sp. D39, Al—treated with 80 µM AlCl_3_, D39 + Al—inoculated with *Cupriavidus* sp. D39 and treated with 80 µM AlCl_3_. Vertical bars show standard errors. Different lowercase letters show significant differences between treatments (least-significant difference test, *p* < 0.05, *n* = 3). The # sign means not detected. DW stands for dry weight.

**Figure 10 plants-12-02334-f010:**
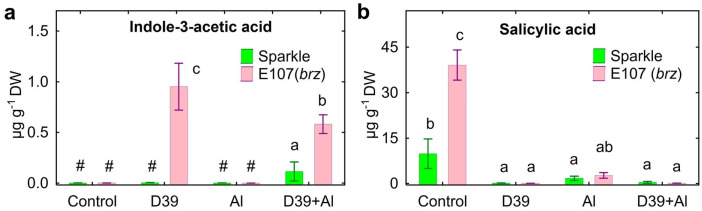
The amount of indole-3-acetic acid (**a**) and salicylic acid (**b**) in the nutrient solution where cultivar Sparkle and E107 (*brz*) mutant were cultivated. Treatments: control—Al-untreated and uninoculated plants, D39—inoculated with *Cupriavidus* sp. D39, Al—treated with 80 µM AlCl_3_, D39 + Al—inoculated with *Cupriavidus* sp. D39 and treated with 80 µM AlCl_3_. Vertical bars show standard errors. Different lowercase letters show significant differences between treatments (least-significant difference test, *p* < 0.05, *n* = 3). The # sign means not detected. DW stands for dry weight.

**Table 1 plants-12-02334-t001:** Growth response of Al-treated cultivar Sparkle and mutant E107 (*brz*) to inoculation with Al-immobilizing rhizobacteria.

Pea Genotype and Treatment	Root		Shoot	
	mg DW Plant^−1^	Increase in Control, %	mg DW Plant^−1^	Increase in Control, %
Experiment 1				
Sparkle				
Uninoculated control	37.0 ± 5.4	-	43.8 ± 5.8	-
*Cupriavidus* sp. D39	56.7 ± 4.1 *	+53	62.2 ± 4.0 *	+42
*Herbaspirillum* sp. E52	39.4 ± 7.3	+6	36.1 ± 7.7	−18
*Paraburkholderia sediminicola* C06	49.1 ± 6.1	+33	47.8 ± 5.8	+9
*Paraburkholderia* sp. A35	50.2 ± 4.1	+36	39.2 ± 3.5	−11
*Rhodotorula* sp. AL1	52.6 ± 4.1 *	+42	49.5 ± 4.8	+13
E107 (*brz*)				
Uninoculated control	9.4 ± 1.2	-	15.7 ± 1.5	-
*Cupriavidus* sp. D39	21.4 ± 2.3 *	+127	19.3 ± 1.2	+23
*Herbaspirillum* sp. E52	6.9 ± 1.1	−27	10.3 ± 0.7 *	−35
*Paraburkholderia sediminicola* C06	12.0 ± 0.7	+28	21.0 ± 2.3 *	+34
*Paraburkholderia* sp. A35	12.6 ± 1.3	+34	23.2 ± 1.4 *	+48
*Rhodotorula* sp. AL1	6.3 ± 0.4	−33	11.8 ± 1.6	−25
Experiment 2				
Sparkle				
Uninoculated control	34.6 ± 2.8	-	39.5 ± 2.5	-
*Cupriavidus* sp. D39	42.2 ± 1.6 *	+22	52.9 ± 2.8 *	+33
*Paraburkholderia sediminicola* C06	44.0 ± 2.5 *	+27	43.3 ± 2.6	+10
*Paraburkholderia* sp. A35	41.3 ± 2.8	+19	42.1 ± 2.3	+7
E107 (*brz*)				
Uninoculated control	22.3 ± 3.0	-	29.1 ± 2.6	-
*Cupriavidus* sp. D39	39.7 ± 7.0 *	+78	45.7 ± 8.1 *	+57
*Paraburkholderia sediminicola* C06	30.0 ± 5.4	+34	38.2 ± 4.9	+31
*Paraburkholderia* sp. A35	32.1 ± 4.0 *	+44	30.3 ± 4.5	+4

All plants were treated with 80 µM AlCl_3_. Data are means ± SE. Asterisks show significant differences between uninoculated controls and the relative treatments for each experiment and pea genotype (least-significant difference test, *p* < 0.05, *n* =10).

**Table 2 plants-12-02334-t002:** Concentration of nutrient elements in roots of cultivar Sparkle and E107 (*brz*) mutant inoculated with *Cupriavidus* sp. D39 and treated with aluminum.

Treatments	Ca (µg g^−1^ DW)	Fe (µg g^−1^ DW)	K (mg g^−1^ DW)	Mg (mg g^−1^ DW)	Mn (µg g^−1^ DW)	P (mg g^−1^ DW)	S (mg g^−1^ DW)	Zn (µg g^−1^ DW)
Sparkle								
Control	884 ± 37 ^a,b^	338 ± 38 ^a,b^	34 ± 1 ^b^	1.4 ± 0.1 ^a,b^	25 ± 1 ^a^	163 ± 6 ^a^	4.8 ± 0.3 ^a,b^	103 ± 14 ^a^
*Cupriavidus* sp. D39	955 ± 116 ^a,b^	316 ± 52 ^a,b^	36 ± 1 ^b,c^	2.2 ± 0.2 ^c^	42 ± 5 ^b^	149 ± 2 ^a^	5.3 ± 0.4 ^b,c^	98 ± 21 ^a^
AlCl_3_	898 ± 44 ^a,b^	439 ± 62 ^a,b^	35 ± 1 ^b,c^	1.2 ± 0.1 ^a^	24 ± 2 ^a^	194 ± 4 ^b^	5.4 ± 0.3 ^b,c^	120 ± 9 ^a^
*Cupriavidus* sp. D39 + AlCl_3_	965 ± 26 ^b^	427 ± 31 ^a,b^	34 ± 2 ^b^	1.3 ± 0.1 ^a,b^	38 ± 4 ^b^	166 ± 6 ^a^	5.3 ± 0.3 ^b,c^	119 ± 9 ^a^
E107 (*brz*)								
Control	884 ± 15 ^a,b^	393 ± 121 ^a,b^	35 ± 3 ^b,c^	1.6 ± 0.1 ^b^	51 ± 8 ^b,c^	197 ± 7 ^b^	5.6 ± 0.1 ^b,c^	162 ± 18 ^b^
*Cupriavidus* sp. D39	973 ± 73 ^a,b^	283 ± 89 ^a^	39 ± 1 ^c^	2.4 ± 0.3 ^c^	66 ± 3 ^c,d^	202 ± 2 ^b^	6.2 ± 0.3 ^c^	161 ± 16 ^b^
AlCl_3_	867 ± 83 ^a,b^	701 ± 122 ^c^	20 ± 3 ^a^	1.2 ± 0.1 ^a^	61 ± 6 ^c^	229 ± 7 ^c^	4.1 ± 0.3 ^a^	169 ± 17 ^b^
*Cupriavidus* sp. D39 + AlCl_3_	804 ± 58 ^a^	565 ± 91 ^b,c^	37 ± 3 ^b,c^	1.7 ± 0.2 ^b^	77 ± 7 ^d^	203 ± 4 ^b^	5.7 ± 0.2 ^b,c^	183 ± 7 ^b^

Data are means ± standard errors. Different lowercase letters show significant differences between treatments (least-significant difference test, *p* < 0.05, *n* = 3). DW stands for dry weight.

**Table 3 plants-12-02334-t003:** Concentration of nutrient elements in shoots of cultivar Sparkle and E107 (*brz*) mutant inoculated with *Cupriavidus* sp. D39 and treated with aluminum.

Treatments	Ca (µg g^−1^ DW)	Fe (µg g^−1^ DW)	K (mg g^−1^ DW)	Mg (mg g^−1^ DW)	Mn (µg g^−1^ DW)	P (mg g^−1^ DW)	S (mg g^−1^ DW)	Zn (µg g^−1^ DW)
**Sparkle**								
Control	808 ± 36 ^b,c^	81 ± 8 ^a^	4.0 ± 0.3 ^a^	2.3 ± 0.2 ^b^	21 ± 2 ^a^	58 ± 21 ^a^	1.8 ± 0.1 ^a,b^	76 ± 3 ^a^
*Cupriavidus* sp. D39	1110 ± 69 ^d^	95 ± 10 ^a^	4.9 ± 0.6 ^a^	2.5 ± 0.3 ^b^	23 ± 2 ^a^	95 ± 4 ^a^	2.1 ± 0.2 ^b^	88 ± 5 ^a,b^
AlCl_3_	744 ± 31 ^b^	93 ± 5 ^a^	4.6 ± 0.4 ^a^	2.2 ± 0.2 ^b^	21 ± 1 ^a^	87 ± 16 ^a^	1.8 ± 0.1 ^a,b^	84 ± 4 ^a,b^
*Cupriavidus* sp. D39 + AlCl_3_	935 ± 29 ^c^	126 ± 18 ^b^	4.9 ± 0.2 ^a^	2.6 ± 0.2 ^b^	24 ± 1 ^a^	97 ± 14 ^a^	2.1 ± 0.1 ^b^	91 ± 2 ^b^
**E107 (*brz*)**								
Control	552 ± 24 ^a^	67 ± 1 ^a^	3.6 ± 0.4 ^a^	2.1 ± 0.1 ^b^	24 ± 2 ^a^	62 ± 23 ^a^	1.5 ± 0.1 ^a^	81 ± 5 ^a,b^
*Cupriavidus* sp. D39	737 ± 47 ^b^	67 ± 10 ^a^	4.2 ± 0.2 ^a^	2.4 ± 0.1 ^b^	29 ± 7 ^a,b^	83 ± 23 ^a^	1.7 ± 0.1 ^a,b^	78 ± 5 ^a,b^
AlCl_3_	424 ± 72 ^a^	88 ± 9 ^a^	4.7 ± 0.7 ^a^	1.7 ± 0.1 ^a^	28 ± 8 ^a,b^	78 ± 23 ^a^	1.3 ± 0.2 ^a^	84 ± 6 ^a,b^
*Cupriavidus* sp. D39 + AlCl_3_	771 ± 79 ^b^	98 ± 2 ^a,b^	4.5 ± 1.0 ^a^	2.3 ± 0.2 ^b^	39 ± 3 ^b^	86 ± 26 ^a^	1.7 ± 0.2 ^a,b^	82 ± 6 ^a,b^

Data are means ± standard errors. Different lowercase letters show significant differences between treatments (least-significant difference test, *p* < 0.05, *n* = 3). DW stands for dry weight.

## Data Availability

The data presented in this study are available on request from the corresponding author. The data are not publicly available due to the rules of the publication activity of the ARRIAM.

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
