# Peer review of "Aluminum-Immobilizing Rhizobacteria Modulate Root Exudation and Nutrient Uptake and Increase Aluminum Tolerance of Pea Mutant E107 (brz)"

_plants, 2023, doi:10.3390/plants12122334_

Round 1
Reviewer 1 Report
plants-2400364.
In this study, the Al-tolerant and Al-immobilizing bacteria Cupriavidus sp. D39, and pea cultivar Sparkle and its Al-sensitive mutant E107 (brz) were used, the plant biomass, exudation of organic acids, amino acids and sugars, etc., was determined. The results are well and interesting.
A minor revision is needed.
1. P2, Line97, “… but indole-3-acetic acid (IAA) was not utilized by Cupriavidus sp. D39”, but in table S1, Cupriavidus sp. D39 could use IAA as carbon source and grow well, please check。
2. P4, Line137, 142, …, “residue” is not clear.
P9, Line276, Figure 10. … “salisylic acid (b)” should be “salicylic acid (b)”.
P12, Line 392- 393, “the obtained results suggest that although Al toxicity induces exudation of organic acids by pea roots, their protective role in Al tolerance of this species is ambiguous.” should be “the obtained results suggested that although Al toxicity induced exudation of organic acids by pea roots, their protective role in Al tolerance of this species was ambiguous.”.
P13 Line403, “The results also suggest that it is difficult…” should be “The results also suggested that it was difficult…”.
P13 Line435, “PGPR use tryptophan” should be “PGPR used tryptophan”.
P13 Line436, “induce expression of genes” should be “induced expression of genes”, …, “[59], increase root exuda-” should be “[59], increased root exuda-”.
P13 Line437, “[60,61] and promote growth of” should be “[60,61] and promoted growth of”.
Author Response
Dear Reviewer!
Thank you very much for valuable comments. We tried to address your comments and answer to your questions. Your comments helped greatly improve the article.
Please, see PDF version for correct line numbers mentioned in this file.
In this study, the Al-tolerant and Al-immobilizing bacteria Cupriavidus sp. D39, and pea cultivar Sparkle and its Al-sensitive mutant E107 (brz) were used, the plant biomass, exudation of organic acids, amino acids and sugars, etc., was determined. The results are well and interesting.
A minor revision is needed.
- P2, Line97, “… but indole-3-acetic acid (IAA) was not utilized by Cupriavidus sp. D39”, but in table S1, Cupriavidus sp. D39 could use IAA as carbon source and grow well, please check。
Response: Thank you very much for this comment. The strain Cupriavidus sp. D39 does not utilize IAA. The Table S1 has been corrected and other data of this Table have been checked.
- P4, Line137, 142, …, “residue” is not clear.
Response: Explanations regarding this term and the method for defining it have been added to the Materials and Methods section (lines 701-705).
Comments on the Quality of English Language
P9, Line276, Figure 10. … “salisylic acid (b)” should be “salicylic acid (b)”.
Response: Correction done (line 329).
P12, Line 392- 393, “the obtained results suggest that although Al toxicity induces exudation of organic acids by pea roots, their protective role in Al tolerance of this species is ambiguous.” should be “the obtained results suggested that although Al toxicity induced exudation of organic acids by pea roots, their protective role in Al tolerance of this species was ambiguous.”.
Response: Correction done (lines 459-461).
P13 Line403, “The results also suggest that it is difficult…” should be “The results also suggested that it was difficult…”.
Response: Correction done (lines 491-493).
P13 Line435, “PGPR use tryptophan” should be “PGPR used tryptophan”.
Response: Correction done (line 523).
P13 Line436, “induce expression of genes” should be “induced expression of genes”, …, “[59], increase root exuda-” should be “[59], increased root exuda-”.
Response: Correction done (line 524).
P13 Line437, “[60,61] and promote growth of” should be “[60,61] and promoted growth of”.
Response: Correction done (lines 524-525).

Reviewer 2 Report
The article presents research about growth promoting bacteria that increase Pea tolerance to Al.
It is recommended to further develop and elaborate the introduction. Some suggestions could be to elaborate on the mechanisms of aluminum tolerance in pea plants and/or related microbes, provide further information about the mutant, and the importance, broader impact or application of this research.
Methods:
It is suggested to be more descriptive and specific when describing the conditions the plants were grown in.
It is suggested to be more descriptive as to how the bacteria were tagged with gfp.
It was noticeable that the mutant has variability in its growth when comparing Experiment 1 and 2.
It is suggested that the figures in Figure 2 all be the same size.
It is suggested to review the use of letters to indicate significant difference in the bars of the figures throughout the article.
There are some grammatical errors throughout the text that need to be revised.
Line 54: investigated in detail.
Line 80: three the more
Line 85: Next,
Line 94: did not produce
Author Response
Dear Reviewer!
Thank you very much for valuable comments. We tried to address your comments and answer to your questions. Your comments helped greatly improve the article.
Please, see PDF version for correct line numbers mentioned in this file.
Comments and Suggestions for Authors
The article presents research about growth promoting bacteria that increase Pea tolerance to Al.
It is recommended to further develop and elaborate the introduction. Some suggestions could be to elaborate on the mechanisms of aluminum tolerance in pea plants and/or related microbes, provide further information about the mutant, and the importance, broader impact or application of this research.
Response: We tried to address your comment and improve Introduction section as you suggested. Additional references have been also added. Please see lines 38-57, 74-80, 88-90, 93-94.
Methods:
It is suggested to be more descriptive and specific when describing the conditions the plants were grown in.
Response: We believe that the plant growth conditions were described in sufficient details and reference No 29 is given for more details. We did it to avoid repetition and self-plagiarism.
It is suggested to be more descriptive as to how the bacteria were tagged with gfp.
Response: Details of how the bacteria were tagged with gfp have been added (Lines 623-624). More details the readers can find in the cited references.
It was noticeable that the mutant has variability in its growth when comparing Experiment 1 and 2.
Response: We agree with this comment. In our experiments variation in plant biomass was probably due to differences in the weight of seeds taken for particular experiment. However, all seeds were calibrated by weight to the nearest 0.1 g and seeds of approximately the same weight were used for each experiment. It should be mentioned that this pleiotropic mutant has poor growth, it is difficult to grow it and to obtain seeds.
It is suggested that the figures in Figure 2 all be the same size.
Response: I understand your remark. However, Figure 2d differs from Figures 2a, 2b, and 2c in the number of columns. If it is reduced, it will be difficult to increase the size of the fonts to fit with other drawings. The size of the Figure 2 itself will practically remain the same, which will not give advantages in the occupied volume of the Figure. Therefore, I ask you to agree to leave this Figure as it is.
It is suggested to review the use of letters to indicate significant difference in the bars of the figures throughout the article.
Response: The letters to indicate significant difference in the bars of the figures have been checked. No mistakes have been found.
Comments on the Quality of English Language
There are some grammatical errors throughout the text that need to be revised.
Line 54: investigated in detail.
Response: Correction done (line 70).
Line 80: three the more
Response: Correction done (line 105).
Line 85: Next,
Response: The sentence seems to be correct. Please, suggest where is the mistake?
Line 94: did not produce
Response: The sentence seems to be correct. Please, suggest where is the mistake?

Reviewer 3 Report
1. Introduction is poorly written, with least significance given to the test plant, PGPR and Al-sensitivity. At least a full parapgaraph of 10-12 lines must be dedicated to each aspect.
2. there are too many figures, apart from 6-7 figures, move others to supplementary data. Also the mean values are not correctly placed in figures 6 to figure 10, please revise those and mention the F and P values on the corner of each graph/figure.
3. In methods section, elaborate, why you choosen these specific plant species, if some previous work is conducted, cite and refer that as well.
4. Conclusion section is poorly written with repetition from abstract, shorten it and write a single paragraph alongwith prospectives.
English language is fine
Author Response
Dear Reviewer!
Thank you very much for valuable comments. We tried to address your comments and answer to your questions. Your comments helped greatly improve the article.
Please, see PDF version for correct line numbers mentioned in this file.
Comments and Suggestions for Authors
- Introduction is poorly written, with least significance given to the test plant, PGPR and Al-sensitivity. At least a full paragraph of 10-12 lines must be dedicated to each aspect.
Response: We tried to address your comment and improve Introduction section as you suggested. Additional references have been also added. Please see lines 38-57, 74-80 and 88-90.
- There are too many figures, apart from 6-7 figures, move others to supplementary data. Also the mean values are not correctly placed in figures 6 to figure 10, please revise those and mention the F and P values on the corner of each graph/figure.
Response: In our opinion, all the presented figures contain very important information that will be of interest to many readers. This is especially true for the new and detailed characteristics of the unique pea mutant E107. Journal rules also allow you to include such a number of Figures in an article. Therefore, we ask you to leave these drawings in the article itself.
The mean values (columns) placed in Figures automatically by the software. Please, clarify what you mean. Perhaps you mean lowercase letters showing significant differences between treatments. Indeed, at the highest columns, the letters are placed near to the columns. This is done so that small columns are also clearly visible when using the optimal size of the ordinate scale. However, the readers can clearly understand which column these letters belong to.
The results were processed by multivariate analysis of variance. What values of F and P (which factor, combinations of factors) you propose to indicate? However, it is very difficult to indicate description of even one of the effects in these Figures, since such additional information will make Figures overloaded (the area of the figure is limited) and difficult to perceive for the reader. Decreasing the font size (for increasing in the area) is also not convenient for the perception of Figures. In our opinion, the Figures provide sufficient statistical information to assess significant differences between the means. Therefore, we ask you allow us do not include information about F and P values into Figures.
- In methods section, elaborate, why you chosen these specific plant species, if some previous work is conducted, cite and refer that as well.
Response: Information about the chosen specific plant species (cv. Sparkle and E107 (brz) mutant) has been added to the Introduction section as suggested by another Reviewer (lines 88-90). So, we propose that there is no need repeating it in the Materials and Methods section. No previous work related to these plant species has been conducted and published by our group. All the related references of other authors about this mutant have been cited.
- Conclusion section is poorly written with repetition from abstract, shorten it and write a single paragraph along with prospective.
Response: The Conclusions have been shortened and combined in a single paragraph as you suggested (lines 719-740).

Round 2
Reviewer 1 Report
no
Author Response
Dear Reviewer!
Thank you very much for review this manuscript.
Reviewer 3 Report
satisfied by revision
Acceptable
Author Response

(The authors gave the same response as above.)
